# PEMFCs Model-Based Fault Diagnosis: A Proposal Based on Virtual and Real Sensors Data Fusion

**DOI:** 10.3390/s23177383

**Published:** 2023-08-24

**Authors:** Eduardo Ariza, Antonio Correcher, Carlos Vargas-Salgado

**Affiliations:** 1Instituto de Automática e Informática Industrial, Universitat Politècnica de València, Camino de Vera s/n, 46022 Valencia, Spain; 2Instituto Ingeniería Energética, Universitat Politècnica de València, Camino de Vera s/n, 46022 Valencia, Spain; carvarsa@upvnet.upv.es

**Keywords:** hydrogen fuel cell, fault diagnosis, proton exchange membrane fuel cell

## Abstract

Proton Exchange Membrane Fuel Cells (PEMFCs) are critical components in renewable hybrid systems, demanding reliable fault diagnosis to ensure optimal performance and prevent costly damages. This study presents a novel model-based fault diagnosis algorithm for commercial hydrogen fuel cells using LabView. Our research focused on power generation and storage using hydrogen fuel cells. The proposed algorithm accurately detects and isolates the most common faults in PEMFCs by combining virtual and real sensor data fusion. The fault diagnosis process began with simulating faults using a validated mathematical model and manipulating selected input signals. A statistical analysis of 12 residues from each fault resulted in a comprehensive fault matrix, capturing the unique fault signatures. The algorithm successfully identified and isolated 14 distinct faults, demonstrating its effectiveness in enhancing reliability and preventing performance deterioration or system shutdown in hydrogen fuel cell-based power generation systems.

## 1. Introduction

Energy transition to renewable sources is necessary in the face of climate change and decreasing non-renewable energy resources. A promising technological option considered as one of the options to store and sustainably use energy is hydrogen [1]. Proton-exchange Membrane Fuel Cells (PEMFCs) are currently one of the most promising technologies for hydrogen utilization as electrochemical devices that convert the chemical energy of hydrogen into electricity [2]. The fuel is oxidized at the anode, releasing electrons, while the oxidant is reduced at the cathode, accepting electrons [3]. PEMFCs [4] have the advantages of low operating temperatures, low weight, high energy density, good response to variable loads, and no pollutant emissions in operation, making them a versatile technology for multiple applications, from stationary generators to small, medium, and large-scale portable units and electric vehicles [4]. However, PEMFCs, which must be fueled with pure hydrogen, must be operated under optimal environmental conditions in order to avoid damage or premature aging. In addition, the complexity of the auxiliary and control systems of the PEMFC modules increases the chances of failure. Several research and development efforts aimed at improving their resistance and reliability, focusing on developing new materials [5], module management, status monitoring, and fault diagnosis [6,7] are currently underway. One recent development in condition monitoring of PEMFC is the integration of a flexible microsensor capable of measuring temperature, humidity, and voltage [8].

Model-based diagnostic techniques use mathematical models to identify and locate faults in the cell [9]. These techniques compare cell operating data with model results to identify any discrepancies. Model-based diagnostic methods help identify faults in specific cell components, such as the membrane or electrodes. Developing better fault diagnosis systems that allow the detection and isolation of any problems occurring in the stack is a work in progress that will help generation systems that include PEMFCs to increase their performance and reliability.

Typical PEMFC failures occur due to poor water management and exposure to high temperatures, corrosive environments, and undesired chemical reactions. Researchers are improving water and thermal management in PEMFCs to prevent flooding and drying [10]. Depending on how the failure can affect the operation of the PEMFC device, cell failures can be classified into transient (recoverable) and permanent (fatal) failures [11]. In a third group are classified the failures related to auxiliary devices such as actuators, sensors, pressure regulators, humidifiers, etc.

Furthermore, failures can be classified into three main types. The first two are related to water management: flooding and dehydration. The third type of failure is membrane deterioration [12]. Both flooding and dehydration phenomena are transient conditions that can be reversed before a significant failure occurs [13].

The water formed inside the cells is usually expelled through the bipolar plates and diffusion channels. When the current increases, water production also increases, forming water droplets that block the passage of protons. Flooding occurs when liquid water is accumulated in the pores of the diffusion layer and the channels, which limits gas exchange between the diffusion channels and between the cathode and anode channels. Flooding occurs mainly on the cathode side due to three leading causes: insufficient evacuation flow rate of produced water, excess inlet water vapor flow, and, finally, condensation of water vapor due to sudden temperature drop. As a result, the active surface area is reduced, increasing current density, increasing ohmic losses and decreasing fuel cell performance [14].

Membrane dehydration occurs when the amount of water vapor produced is insufficient for a prolonged period of operation. In a PEMFC, proton transfer is ensured by the water molecules contained in the membrane. Dehydration of the membrane increases the electrical resistivity of the cell, limits the maximum current density, and reduces the thickness of the membrane. The charge current must be increased, and the gas flux reduced, to solve this issue [15].

Membrane deterioration failures are cumulative conditions produced over time. This type of failure is considered a permanent and irreversible failure where membrane parameters are affected [16]. The diffusion constants are significantly altered in most cases and the pressure gradient between the anode and cathode channels drops. Additionally, drastic changes in the loading regime produce abrupt changes in the internal pressure that eventually lead to mechanical fatigue of the membranes and their subsequent rupture [17]. The leading causes of cell degradation are load regime changes (56.5%), starting and stopping process (33.0%), overloading (5.8%), and idling (4.7%) [18]. Additionally, the manufacturer, Ballard (Vancouver, Canada), reports that, during long storage periods, the cell’s performance can reduce by about 10% per year [19].

The fault diagnosis process consists of three activities: fault detection, fault isolation or identification, and fault analysis [20]. Fault detection occurs during system operation and consists of discovering the occurrence of the fault. When the fault is detected, fault isolation can be initiated, i.e., identifying where the fault has occurred. Finally, failure analysis determines the failure’s magnitude, nature, or cause.

Petrone et al. [21] groups fault diagnosis methods in PEMFC devices into model-free and model-based methods. Model-free methods allow detection and identification of the fault through human experience and knowledge and in the use of techniques based on analyzing input and output data. In model-based methods, the fault diagnosis is mainly performed by comparing a real signal and a modeled one. Their difference or residual allows us to presume the occurrence of a fault. Therefore, this method is also known as residual-based diagnosis.

A more recent proposal [22] classifies, in more detail, the methods used in PEMFC fault diagnosis, creating three groups: data-based methods, experimental test methods, and model-based methods. Experimental test methods, in turn, can be classified into failure operation methods and magnetic field methods. Data-based methods are subdivided into signal processing methods and pattern classification methods. Finally, model-based methods are classified into black-box and analytical models (Figure 1).

Within the research efforts related to fault diagnosis in PEMFC, ref. [23] makes an approach to the diagnosis of flooding and dehydration of the cells based on signal processing and pattern recognition, where the voltage signal is processed by a Fast Fourier Transform (FTT) to calculate the total harmonic distortion (TDH) that increases the amplitude of the harmonics in case of drying and decreases them in case of flooding. Identification with supervised and unsupervised classification methods achieved classifications from 84% to 98.5%.

Another graphical approach for PEMFC cell failure diagnosis is proposed by [24] that analyzes the one-dimensional (1D) voltage data of a single cell and converts it into a two-dimensional (2D) image; optimal characteristics are determined by Fisher Discriminant Analysis (FDA). The K-means clustering method analyzes the data collected from two different single cells in flooding and dehydration failure states. Wavelet transform was used for the analysis.

Mixing several techniques [25] employs an equivalent circuit model that was identified by combining a genetic algorithm (GA) and the Levenberg–Marquardt (LM) algorithm. For fault diagnosis, a method based on Adaptive Neural Fuzzy Inference System (ANFIS) and Electrochemical Impedance Spectroscopy (EIS) was applied to a 12-cell 196 cm2 stack to which the stoichiometric ratio and air humidity would be varied to produce membrane flooding and drying. Most of these techniques have been applied to diagnose membrane flooding or membrane dehydration failures applied to a cell or stack. However, a PEMFC module is not only an array of cells. It is a device that integrates a series of equipment that allows the autonomous operation of the module.

Under the PEMFC module concept and with a model-based approach, ref. [26] performed process estimation, waste generation, and a hierarchical method for FDD detection and diagnosis that creates a multi-stage structure. In the first stage, faults were diagnosed at the subsystem level and then at the component level. Residuals were used as fault indicators, considering various abrupt and degradation faults. Experimental results verified the accuracy of the model-based approach and demonstrated that the proposed multi-stage hierarchical method effectively diagnosed faults in a system. Failures were detected when the residuals were outside the limits set for regular operation.

Finally, cutting-edge fault diagnosis integrates real-time data via advanced algorithms, surpassing model-based methods. It adapts dynamically, enhancing accuracy and timeliness in identifying and addressing faults across complex systems. In their study, ref. [27] developed a convolutional neural network (CNN) model. The system accurately identifies particle parameters based on PLDD-TENG output signals, even distinguishing particle types within mixed solutions. This innovation culminates in an intelligent visualization system enabling real-time monitoring of sediment particles, with significant implications for understanding the triboelectric effect in two-phase flow dynamics.

According to the literature review, it was found that there is a primary group of research works on PEMFC fault diagnosis, which encompasses the majority of publications. In this group, fault diagnosis techniques are focused on understanding the effects of fault conditions on individual cells, particularly those related to cell dehydration or flooding. These techniques are applied to a single cell and are developed under controlled laboratory conditions, where temperature and pressures are managed. Some notable works in this area include: [6,7,8,13,14,15,16,17,24,25].

A second group of research works investigates models applied to individual cells (or stacks of cells). Similar to the previous group, these studies aim to establish the impacts of improper operating conditions. To achieve this, experiments are conducted under controlled temperature and pressure conditions, and the stack’s operation is simulated to estimate voltage response to a current profile. In this context, PEMFC models are primarily of the black-box or equivalent circuit type [9,11,12,18,20,21,23]. Up until this point, fault diagnosis is performed by monitoring signals and seeking the replication of the previously induced fault. Published works on PEMFC diagnostic algorithms rely on expert knowledge and apply techniques such as fault hierarchy, wherein subsystems isolate a detected fault and then by components without using models. These works serve as diagnostic guidelines [26].

Therefore, specific issues can be concluded after studying the literature for PEMFC fault diagnosis. First, the operating temperature of the cells conditions their electrical performance and cell life, creating situations of flooding or dehydration of the membranes. Moreover, the diagnostic techniques based on models apply to a more significant number of PEMFC devices. In general terms, the greater the complexity of the model, the greater the capacity for fault diagnosis and condition monitoring, being the analytical model’s valuable tools. In addition, most models study the electrical behavior of the cell, i.e., the relationship between current as the input signal and voltage as the output signal, taking temperature and gas pressures as controlled variables under laboratory conditions. Therefore, an improvement in the diagnosis of PEMFC failures requires a model that considers the operating temperature and allows the simulation of various operating conditions and failures without resorting to tests that reduce the useful life of the PEMFC module.

The fault algorithm proposed in this study is based on analytical models and observers. The employed model considers the PEMFC module’s actual behavior, and its block design provides a wide array of signals. The integration of the model and the diagnostic algorithm distinguishes itself from current methods in the following ways: (a) The use of a complex analytical model with 12 observers or residuals, (b) The segmentation of signals into different operational zones of the module, (c) Statistical analysis of data to establish the significance of the fault relative to the median and the standard deviation values of the residuals, (d) The utilization of a fault matrix weighted with nine possible classifications of the fault signature—four positive values, four negative values, and zero.

This work’s scientific contribution lies in identifying and discussing specific issues related to PEMFC fault diagnosis, which are essential for advancing the field and improving the reliability of hydrogen fuel cell-based power generation systems. Although all the results are applied to a specific PEMFC, the methodology can be used to develop failure diagnosis for cells with the same characteristics. The equations representing the physical phenomena occurring within the stack are grouped into blocks. Such blocks utilize both real and calculated input and output signals. The calculated signals serve as virtual sensors that generate non-measured signals. Real or virtual measured signals are compared with the digital twin simulation to generate the residuals. With more observers, the characterization or signature of faults becomes more detailed, enabling more accurate fault isolation. A virtual sensor’s accuracy depends on the precision of the identification process, so it is always preferred to have real measured data and create observers only for signals that can not be measured or are very expensive to measure. The proposed technique uses virtual sensors to avoid including extra sensors in the cell, allowing the direct use of the proposed diagnoser in commercial PEMFCs.

The organization of the paper is as follows. After this introduction, Section 2 describes the fuel cell and its model, the model identification process, and the faults considered in this study. The results are presented in Section 3, which includes a deep statistical study of the residuals and the development of the failure signatures based on the residual medians and the standard deviation (SD). Finally, discussion and future work are addressed in Section 4 and Section 5.

## 2. Materials and Methods

### 2.1. The PEMFC Power Module

The tested fuel cell module is a *NEXA 1.2 kW* Ballard (Vancouver, BC, Canada) unit, installed in the Distributed Energy Resources Laboratory (LabDER [28]) at Universitat Politècnica de València (Valencia, Spain) (UPV). Table 1 shows the datasheet of the Module [29].

The Nexa fuel cell integrates all the necessary accessories for its operation, including an automatic inlet valve and a pressure regulator. The air required for the reaction is taken from the environment, filtered, and compressed by the device. The flow is measured through a mass flow meter. The speed regulation of the fan is used to regulate the temperature of the module, which is also measured. The stack voltage is measured for the whole cell assembly, so it is possible to determine if there are hydrogen residues and remove them with the accumulated impurities when the purge valve is opened.

As a protective measure, if the current produced by the fuel cell exceeds the maximum value, the controller opens a relay and acts on the air compressor to maintain the correct stoichiometric ratio. Nexa’s subsystems include a hydrogen regulation system and an automatic system that performs the device’s control, monitoring, and safety functions. The fuel cell operation is subdivided into the following subsystems [29].

#### 2.1.1. Hydrogen Subsystem

The Nexa module requires a supply of pure and dry hydrogen. The module’s hydrogen regulator allows the fuel to remain pressurized inside the cells while the hydrogen is consumed. The subsystem includes the following components: a pressure transducer to monitor and guarantee the hydrogen supply, a pressure relief valve, a solenoid valve to shut off the hydrogen supply on device shutdown, a pressure regulator, and a hydrogen level detector to detect fuel leaks.

#### 2.1.2. Oxidizing Air Subsystem

The Nexa module is supplied by excess oxidizing air generated by a compressor that draws air from the environment and filters it. The air passes through a moisture exchanger that takes advantage of the humidity produced at the device cathode. A mass flow sensor detects the amount of air entering the stack and allows regulation of the compressor speed to match the demanded current. Excess water production is discharged from the module as liquid and vapor in the air exhaust.

#### 2.1.3. Cooling Subsystem

To maintain the module’s operating temperature close to 65 °C, a sensor located in the cathode exhaust measures the temperature. With this information, a fan installed at the fuel cell’s base is controlled and blows air through the stack’s vertical cooling channels. During normal operating conditions, small amounts of hydrogen are occasionally released into the cooling system to purge the system. A hydrogen sensor at the cathode exhaust provides hydrogen concentration information to keep fuel levels below the lower flammability limit (4% hydrogen by volume).

#### 2.1.4. Electronic Control System

A digital analog control system operates the Nexa fuel cell with a sampling time of 200 ms. The system uses input signals from the various sensors and commands the hydrogen valve and purge valve, the air supply compressor speed, the cooling system fan speed, and the load relay. When the operating variables reach the safety limits, alarms are triggered, and the shutdown protocol of the device can be activated. Some alarms, such as hydrogen leakage, failures in the test system, and software faults, are considered non-resettable and do not allow the start-up of the fuel cell until the intervention of specialized personnel.

### 2.2. PEMFC Model

A detailed model of the Nexa fuel cell is presented in [30]. The model was fitted with measurements from different Nexa operations periods. In addition to manufacturer information, measured data included the stack temperature and voltage, pressures, current, and intake airflow.

The model was developed using LabView, and it consists of several equation blocks regarding different features such as voltage losses, heat losses, active pressure, and overall potential. Each block allows the estimation of non-measured variables, such as the reaction’s temperature, the voltage of the terminal, or inner pressures. Figure 2 shows a scheme of the model and the coefficients of each equation block. Figure 3 shows the results of the fitting process. In Figure 3a, real and simulated voltage signals are represented, and in Figure 3b, temperature signals are shown. The model’s coefficients were fitted using a Scout Evolutionary Algorithm and achieved an error rate of 2.21% for the voltage signal and 1.97% for the temperature signal.

The complete list of signals of the model is shown in Table 2. The signals have been classified as:Inputs: input signals provided by the Nexa module.Virtual: model calculated signals.Outputs: output signals provided by the Nexa Module.

**Table 2 sensors-23-07383-t002:** Nexa 1.2 kW fuel cell model variables.

Signal	Classification	Description
TInitial	Input	Initial temperature of the cell
TRoom	Input	Temperature of the cell environment
I	Input	Cell demand current
AFlow	Input	Airflow
PAnode	Input	Anode pressure
PCathode	Input	Cathode pressure
P_H_2_	Virtual	Hydrogen partial pressure
P_O_2_	Virtual	Oxygen partial pressure
Act_1	Virtual	Activation voltage drops
Act_2	Virtual	Activation voltage drops
Conc	Virtual	Concentration voltage drops
Omh	Virtual	Ohmic voltage drops
VCell	Virtual	Individual cell voltage
ΔG	Virtual	Gibbs free energy
TReaction	Virtual	Reaction temperature
TLoss	Virtual	Temperature loss
Vout	Virtual	Terminal output voltage
VStack	Output	Stack output voltage
TStack	Output	Stack temperature

The model utilizes six signals measured by the fuel cell stack’s sensors. It measures the output voltage and temperature. Aside from being used as an output signal, the temperature also serves as an input signal. While this increases the model’s complexity, it allows for estimating the module’s thermal behavior. The model employs 47 equations to calculate phenomena such as gas diffusion, partial pressures, Gibbs free energy, cell potential, conservation of matter, and thermodynamic energy balance, among others. These equations are adjusted using 16 coefficients. Among them, four are associated with partial gas pressures, four with voltage drops, two with cell potential, three with thermal dynamics, two with temperature and voltage inertia, and finally, one with the overall module behavior. Given the model’s nonlinearity and the intricate interaction of the equations, parameter adjustment is performed using evolutionary optimization algorithms.

#### Model Validation and Operating Modes

In the Nexa PEMFC, values such as internal pressures, reaction temperature, or voltages, among others, change non-linearly depending on the temperature and the current. For this reason, it is necessary to establish differences in the operating current ranges of the module. These ranges are also related to the conditions with more significant cell degradation [18]. Specifically, in the Nexa PEMFC, the operating zones have been divided depending on the range of the demand current and its trend into four zones: idle, load ramp, discharge ramp, and load regime change. Figure 4 shows real data current demand curves applied to the fuel cell.

The model described in Section 2.2 (fitted as explained in [30]) has been validated with different real current data profiles (shown in Figure 4). Considering the two main outputs (voltage and temperature), the overall mean error is lower than 2.5%. Therefore, it is concluded that the model can be used as a digital twin of the Nexa PEMFC.

### 2.3. PEMFC Faults

The NEXA fuel cell issues two types of alerts, warnings and fault alerts, which differ in the sensor measurement range. Warning alerts indicate potential issues that require attention but do not disrupt immediate operation. Fault alerts signify critical malfunctions that demand immediate action to prevent system failure. When the controller detects a warning alert, it modifies the module’s operating conditions, while a failure alert forces the module to shut down. On the other hand, alerts can be resettable or non-resettable. Non-resettable faults present safety risks. Appendix A shows the list of Nexa module alerts according to the reading levels, and the possible causes have been included. Appendix A provides guidelines on possible abnormal operating situations of the Nexa fuel cell. It shows a list of alerts associated to some symptoms and their possible causes, offering procedures to improve the safety and performance conditions of the module, and constituting a basis for fault diagnosis tasks. However, it is necessary to deepen the activities related to the detection, isolation, and identification of faults in the PEMFC.

The list of considered faults has been created considering the fuel cell operation manual (see Appendix A) and the most common failures described in the literature [31]. Because of the cost of the fuel cell and the conservative behavior of the controller, it is not possible to physically generate failures in the system. Therefore, all faults are simulated in the digital twin validated in the previous section.

The faults are simulated by modifying input signals or parameters of one of the equation blocks. As a result, the signals of the digital twin are altered. A fault-free model runs in parallel and allows the residual generation for all the available signals. In this case, the residuals are the point-to-point difference between the digital twin’s output signals and the fault-free model’s output signals. Table 3 summarizes the faults and the signals that must be changed to conduct simulations.

## 3. Fault Diagnosis Process

One of the most used model-based diagnosis techniques is the residual analysis. A fault-free model runs in parallel with the real system, and the signals from both systems are compared. Differences between real and simulated signals are called residuals and can be used to detect abnormal behaviors such as aging or faults.

Model-based observers can be used as virtual sensors based on analytical models for fault diagnosis and condition monitoring. Depending on the model’s level of detail, it is possible to perform the diagnosis in real operating conditions, obtaining the possibility to perform online diagnostics. The block design of the PEMFC model and its level of detail allow for the diagnosis using both real and observed output signals to compute the residuals.

The detection of failures, understood as the activation of residuals, starts with verifying the statistical normality of the data. However, thanks to the central limit theorem and the large sample size (*n* > 50), the non-normality of the data can be hindered. Otherwise, applying the Kolmogorov–Smirnov test with the Lilliefors correction, developed as an alternative to the Shapiro–Wilk test, is necessary.

The Kolmogorov–Smirnov and Shapiro–Wilk [32] tests are statistical methods used to assess the normality of a dataset. The Kolmogorov–Smirnov test compares the cumulative distribution of the data with a theoretical normal distribution, yielding a single statistic. The Shapiro–Wilk test calculates a statistic based on the correlation between the data and the expected normal distribution. Both tests provide a *p*-value that indicates whether the data significantly deviates from a normal distribution. A low *p*-value suggests non-normality. While the Kolmogorov–Smirnov test is sensitive to deviations throughout the distribution (it is used for *n* ≥ 50 samples), the Shapiro–Wilk test is particularly effective for smaller sample sizes (*n* < 50 samples).

The second step is to determine statistically whether the residuals are affected by the failure or are due to modeling errors, for which the *t*-Student test is applied with a significance level of 0.5%. It is also necessary to determine whether there is a significant difference between the residuals of the signals. This issue is resolved by applying a one-way ANOVA test that can be parametric or non-parametric, i.e., to define whether the residuals have equal variance. It is defined by Lenvene’s homoscedasticity test of the data. In this work, the non-parametric Kruskal–Wallis one-way ANOVA test is applied. The Kruskal–Wallis [33] test is a non-parametric statistical test used to determine if there are significant differences in the medians of three or more independent groups. It is an extension of the one-way ANOVA for situations where data does not meet the assumptions of normality or homogeneity of variances.

The next step is to analyze which residuals have significant differences. This analysis is done with the Dwass–Steel–Critchlow–Fligner (DSCF) test, which compares the residuals pairwise. It is tuned using the Wilcoxon (W) statistic with a *p*-value of 5%. The DSCF test [34] is a non-parametric multiple comparison test used to assess differences between multiple groups in statistical data. It is particularly useful when data violate assumptions of normality and homogeneity. DSCF extends the pairwise Wilcoxon rank-sum test to provide adjusted *p*-values for multiple comparisons while controlling the familywise error rate.

In the tests performed, the data are non-homoscedastic. Therefore, the median statistic is used as an initial tool for fault isolation. The standard deviation is another positive residual comparison statistic that can be used in fault isolation.

### 3.1. Data Analysis Example

The documents published by the module manufacturer consistently highlight a recurrent current sensor calibration failure (F7). This specific fault can potentially cause erroneous current signals, either higher or lower than the actual values, consequently affecting the oxidant and ventilation supply of the module and, in turn, impacting its temperature. This section presents a comprehensive simulation and analysis of the F7 fault scenario.

A real current demand profile has been used as an input of the digital twin (Figure 4b). This profile includes three of the four fuel cell operating zones mentioned in the previous section: idle with minimum current levels (time = [0, 100]), a loading ramp where the current increases (time = [1000, 2000]), and load regime changes (time = [2000, 2547]). Obtaining real data in overload conditions is not possible due to the performance of the fuel cell protection systems. Figure 4b shows the input current profile, and Figure 5a,b show the simulated output signals of voltage (V) and temperature (T) when fault F7 is active.

Figure 6 shows that the resulting residuals are correlated to reducing the voltage and temperature signals, creating positive residuals for the outputs of gas partial pressures and temperatures. The output signals related to internal resistances and voltages generate negative residuals. It can be observed that, in general terms, all signals follow the dynamics of the temperature or voltage signals.

Residuals never reach zero values because of modeling errors or signal noise. Moreover, for fault detection purposes, data must be brought to comparable values between residuals, i.e., it is needed to relativize or normalize the residuals point to point. Therefore, each residual data is divided by the real signal data, so the relative residual data *r_k_* is calculated employing the following equation:(1)$$r_{k}=\frac{s_{r}-s_{s}}{s_{r}}$$
where *s_r_* is the signal measured from the real system, and *s_s_* is the simulated data.

Then, the data for each fault and each residual are divided into the operation zones under the nomenclature $F_{i}{\_Z}_{j}$, where $F_{i}$ identify the fault (i = [1, 2, …, 14]) and $Z_{j}$ represents the operation zone (idling—*Z*_1_, load ramp—*Z*_2_, load regime variation—*Z*_3_).

The statistical analysis of the data must be carried out independently for each operating zone to establish whether the residues are due to the induced failure and not only to the stochasticity of the process. The application of statistical tests allows proving if there are significant statistical differences between the residues or if all the residues were affected in the same magnitude.

This section also shows the data analysis process applied to the first operating zone, idling, of the first fault (*F*_7__*Z*_1_). The rest of the faults and operation zones in each fault show the graphical result, the median tables for all faults, and operation zones.

For failure detection, the first step is the normality check, which was performed using the Kolmogorov–Smirnov test with the Lilliefors correction, developed as an alternative to the Shapiro–Wilk test for normality analysis when the sample size is greater than 50. Table 4 shows the test results for *Z*_1_. The rows contain each of the 12 residual signals. W represents the Shapiro–Wilk statistical operator, and *p* represents the *p*-value.

Since the *p*-value in each of the residuals is less than 0.05, it is concluded that the residual signals do not have a normal distribution. Despite this and given that, by the central limit theorem, the non-normality of the data for large samples can be hindered, it is resorted to the *t*-Student test.

Table 5 presents the application of the *t*-test to each residual, where the second column (*t*) represents the t-statistic, Dof represents the degrees of freedom equivalent to the number of data minus one (N − 1), and *p* is the *p*-value.

Based on the obtained results, the failure significantly impacts all the residuals. None of them exhibit a mean of zero within the 5% significance level.

The next step is the isolation of the failure. This process can be approached by establishing which residuals the failure affects. From a statistical point of view, a one-way ANOVA test is helpful for this analysis. Nevertheless, before the test can be implemented (as it can be parametric or non-parametric), the homoscedasticity of the data must be established using the Lenvene test. Table 6 shows the results of this test. The residuals are not equal variances because the *p*-value is less than the 5% significance level. In Table 6, F is Fisher’s test statistic, which is based on the ratio of the sum of squares of the residuals, Dof represents the degrees of freedom of the residuals, and *p* represents the *p*-value.

According to the Levene test results, the ANOVA test to be applied is the Kruskal–Wallis one-way non-parametric ANOVA, which allows for determining if there are significant differences between the residual signals. Table 7 shows the signal analyzed, the Chi-square, the degrees of freedom, and the *p*-value.

As presented in Table 7, the *p*-value, being less than 5%, confirms a significant difference among the residual signals. This indicates that at least one residual significantly deviates from the others. To investigate these differences further, a pairwise analysis of the residuals is conducted using the Dwass–Steel–Critchlow–Fligner test. The results of the pairwise comparison, performed with a *p*-value of 5%, are shown in Table 8, employing the Wilcoxon (W) statistic.

*p*-values equal to 1.0 indicate that the medians of the residuals are identical, for example, in the pair (1, 2). *p*-values greater than 0.05 show that the medians are similar, for example, the pair (3, 6). In these cases, it is impossible to distinguish the failure’s effect in these residuals. *p*-values less than 0.05 indicate a difference in the medians compared, verifying the usefulness of the medians of the residuals as an initial tool for fault isolation.

The medians of each residual are shown in Table 9, which also shows the mean and the standard deviation.

Figure 7 graphically shows the results of Table 9 using the box-and-whisker plot. It is observed that the signal of residue 5 is the most affected by the failure; residues 5 and 6 are also remarkable for their value but not for their dispersion. The test shows that some pairs are indistinguishable ((1, 2) and (7, 11)). Therefore, residues 3, 4, 5, 6, 8, 9, 10, and 12 are isolable.

The previous procedure is repeated for each of the zones of operation of each failure as an extension of the example that has been presented. Figure 8 shows the graphical result for *F*_7__*Z*_2_, where the general behavior of the residuals is similar to the previous one but with greater dispersion and magnitude. Residuals are isolable except for pairs (1, 2) and (7, 11). Being remarkable, residual 5 changes its sign and increases its dispersion concerning residual 6.

### 3.2. Failure Signature and Failure Identification

The previous section illustrates the procedure to determine which residuals can be used for fault identification. The specific combination of residuals for each failure and their values define the fault signature for each failure. The failure signature from the medians can be built, taking into account the size of the median, so the residual will not be binary. Each residual median can be classified according to Table 10. Medians falling within the (−0.05, 0.05) range were assigned a “0”. This range replaces values that are less than ±5%, corresponding to twice the magnitude of the modeling error.

Table 11 shows the failure signature matrix for failure F7. Besides the median quartile for each residual, Ø_Z_ makes a horizontal count of the weighted residuals different from “0”, and Σ_r_ is the vertical summation of the values of the weighted residuals of the operating zones of each fault. The Z4 values of the residuals have not been considered due to the absence of data.

The complete median residual matrix failure signature is shown in Appendix B. This table has to be analyzed to study which failures can be isolated. An easy way to look for different failure signatures is by looking at Ø_Z_ and Σ_r_. The following cases are presented:Case 1: Ø_Z_ in a zone is different from any other fault. This is the case, for example, of F10 at zone 3. This failure could be identified when the PEMFC is working in that zone.Case 2: Ø_Z_ in a zone is equal to another fault, but Σ_r_ is not. This is the case, for example, of failures F7 and F8. These failures could be isolated because both residual signatures are different.Case 3: Ø_Z_ and Σ_r_ values are equal in all the zones, as in faults F4, F5, F6, F9, F12, and F14. These failures could not be isolated, making it necessary to use other fault characteristics for the diagnosis process.

With this procedure, it can be evidenced that failures F1, F2, F3, F7, F8, F10, F11, and F13 present different marks to those of the most residuals, making them isolable. However, failures F4, F5, F6, F9, F12, and F14, cannot be distinguished, necessitating using another characteristic for the isolation.

A similar treatment to that applied to the medians of the failures was employed with the standard deviation as another helpful feature in the isolation of the failure. Table 12 shows the part of the matrix related to F7. The complete matrix can be seen in Appendix C. It can be seen that any fault has a unique fault signature when combining information from the weighted median and the standard deviation.

### 3.3. Fault Diagnosis Algorithm

The algorithm to monitor the PEMFC and perform failure detection and isolation is shown in Figure 9. The algorithm starts with failure detection by monitoring any residual deviation. The residuals’ median and standard deviation are calculated in the second step. In the third step, it is checked if the calculated medians exceed the detection threshold. If this condition is not met, the system continues monitoring the PEMFC module. If the condition is completed, the fault signature is created by weighting the medians and SD in the fourth step. The fifth step compares the median fault signature with the fault matrix. If the fault can be isolated, the process ends with some recommendations. Otherwise, the SD failure signature is compared with the SD failure matrix in the sixth step. Positive fault isolation leads to the recommendations table. Otherwise, it is understood that it is an unknown fault, and this is characterized. The failure matrix is updated in the eighth step, and the system returns to the fifth step.

The intervention actions are a guide provided to the operator to facilitate the inspection of the module and, if necessary, to remedy the situation. The faults detected and the suggested actions are listed in Table 13 below.

## 4. Discussion

Traditionally, significant advances in PEMFC fault diagnosis and condition monitoring have focused on faults arising from water management, i.e., detecting membrane flooding or dehydration. However, there are few comprehensive approaches to PEMFC module fault diagnosis. The PEMFC fault diagnosis process developed in this paper applies to the entire Nexa 1.2 kW PEMFC module and, if adjustments are made, readily applies to many other commercial modules.

The work presented integrates the application of many techniques that generate advances for improving PEMFC technology. To reach this point, it was necessary to refer to a previous complex work that included the development of an analytical model of a PEMPC that takes into account the temperature as an output and feedback signal, the study and application of different optimization algorithms for the parametric adjustment of the model, and, finally, the creation of this proposal for fault diagnosis for PEMFC.

Since the model identification process can be performed periodically at a low computational cost, the developed tools can complement other fault diagnosis techniques, such as parametric identification. Another technique that can be integrated into the diagnosis is the periodic characterization of the stack using the module polarization curve because the output voltage can be easily contrasted, evidencing the voltage drops of the module that are symptoms of internal phenomena of obstruction or degradation of the membrane.

A model that behaves as a digital twin allows multiple nondestructive tests. However, the diagnosis has an uncertainty inherent to an analytical model limited to the available signals and the fitting process. Including new sensors in the PEMFC and the availability of real signals to validate the virtual sensors listed in Table 2, will contribute to reducing this uncertainty. Therefore, it will be interesting to include pressure sensors for P_H_2_ and P_O_2_ signals, measuring the differential pressure between different points of the diffusion channels of the cells. More quickly, voltage sensors could be added to measure the behavior of the voltages of each of the cells as a function of the current and to determine the values of voltage drops represented by: Act_1, Act_2, Conc, and Ohm. Temperature sensors are easy to include in the commercial Module to measure gains and heat losses for different points of the stack. These sensors would provide information to contrast the signals Tloss and TReaction. Nevertheless, it is only possible to measure the dG signal indirectly due to the characteristics of the units. The installation of all these sensors could be approached in two ways: as a supervisory system external to the PEMFC whose connection does not interfere with the control and safety board of the module or by integrating the sensors within the fault diagnosis and status monitoring software on the module control board, the latter option being the ideal alternative.

## 5. Conclusions

This paper presents a model-based fault diagnosis method for fuel cells developed in LabView. Based on expert knowledge of the Nexa fuel cell, its operation was divided into four modes: idling, load increase, speed changes, and load reduction. Then, with the modification of the model input signals, 14 failures were simulated, and thanks to the model design, it was possible to extract 12 output signals that were compared using residual techniques. Output signals from the modules are not measured in the commercial version of the fuel cell, but they can be estimated by models acting as virtual sensors. Comparison between real and measured data (with real or virtual sensors) generates a set of residuals.

The residual data were analyzed with statistical methods to characterize them and thus create a fault signature. The analysis of the data, separated into zones or modes of operation, evidenced: that the data do not obey a normal distribution, that the non-zero residuals are the result of the fault simulation, and that there are significant differences between the medians of the residuals. This process was repeated for the three zones of operation of the 14 faults, for a total of 42 data sets.

From the data analysis, it was possible to create a signature for faults: F1, F2, F3, F7, F8, F10, F11, and F13. In a second step, a weighting of the standard deviations of the 42 data sets was done, thus isolating the rest of the faults: F4, F7, F6, F9, F12, and F14. With these tools, a combined fault matrix was consolidated. The matrix was validated with a second data set. Still, with a load profile that includes a new module operation zone containing a decreasing current ramp and with it, the matrix was updated to obtain four operation zones. With the application of the diagnostic algorithm, the fault matrix was updated, and two other load profiles were validated, obtaining results that conclude the fault diagnosis process is valid. The time required for analyzing the residuals of this second load profile with 1500 s was 14 ms, using an AMD Ryzen 5 5500U processor.

According to the conducted tests, the most likely failures on the PEMFC are related to temperature and water within the module, meaning membrane flooding or dehydration. In the second place, membrane-related failures can be mentioned, either due to degradation or cracks. However, common failures result from a direct fault in some devices, such as the fan, pressure regulators, gas flow sensors, etc.

The importance of this work lies in identifying and discussing specific issues related to PEMFC fault diagnosis, which are essential for advancing the field and improving the reliability of hydrogen fuel cell-based power generation systems. Although all the results are applied to a specific PEMFC, the methodology can be used to develop failure diagnosis for cells of the same characteristics.

Future efforts will concentrate on enhancing the analytical model through the modularization of equations to encapsulate clusters of cells. This method will facilitate the individual assessment of cell degradation, enabling the generation of notifications for implementing rejuvenation strategies or partial stack replacements. Another avenue of development involves the incorporation of sensors within the Module, augmenting the count of measurable signals accessible for an extended diversification of fault categorization within the fault matrix.

## Figures and Tables

**Figure 1 sensors-23-07383-f001:**
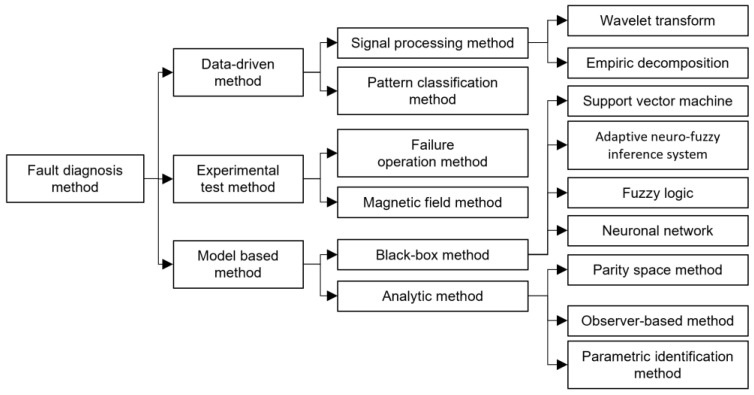
Classification of diagnostic methods applied to PEMFC devices [22].

**Figure 2 sensors-23-07383-f002:**
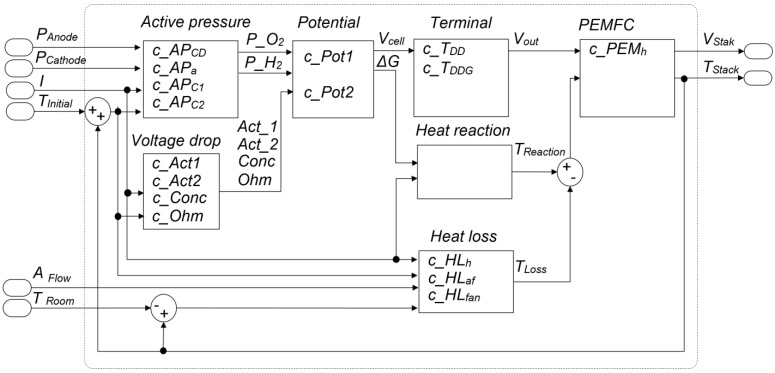
Model of the Nexa fuel cell and its coefficients.

**Figure 3 sensors-23-07383-f003:**
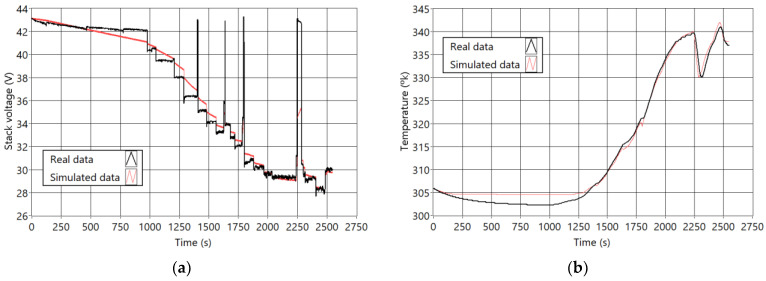
Comparison of the used real and simulated signal. (**a**) voltage signals; (**b**) temperature signals.

**Figure 4 sensors-23-07383-f004:**
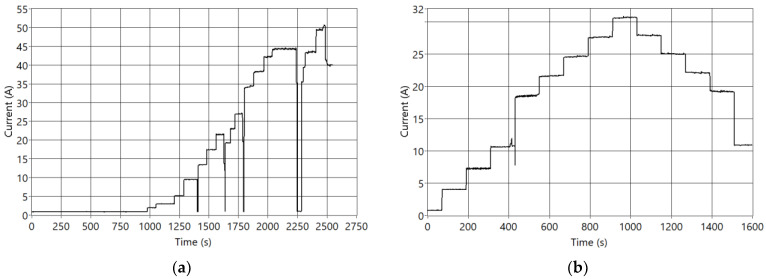
Real current demand curves. (**a**) Idle, load ramp, and regime change; (**b**) Load and discharge ramps.

**Figure 5 sensors-23-07383-f005:**
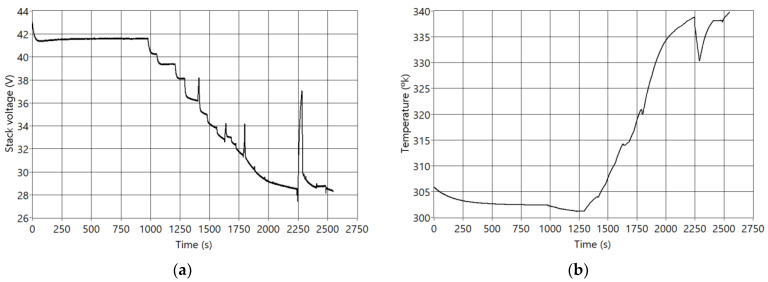
System outputs when the fault F7 is active. (**a**) Stack voltage; (**b**) Stack temperature.

**Figure 6 sensors-23-07383-f006:**
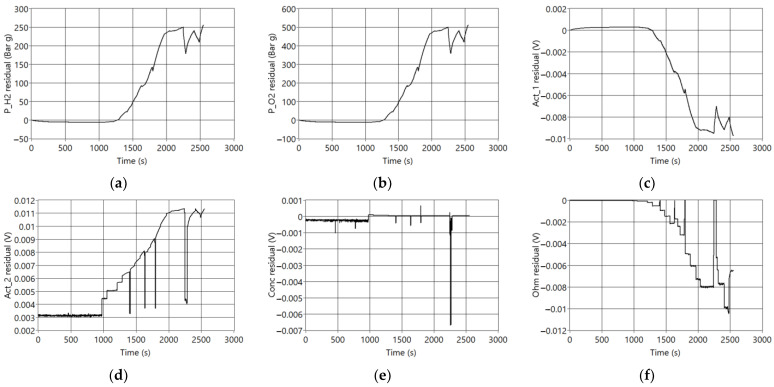
Residuals generated when F7 is active. (**a**) P_H2 residual (Bar g); (**b**) P_O2 residual (Bar g); (**c**) Act_1 residual (V); (**d**) Act_2 residual (V); (**e**) Conc Residual (V); (**f**) Ohm residual (V); (**g**) Vcell residual (V); (**h**) ΔG residual (Jul/mol); (**i**) TReaction residual (K); (**j**) TLoss residual (K); (**k**) VStack residual (V); (**l**) TStack residual (K).

**Figure 7 sensors-23-07383-f007:**
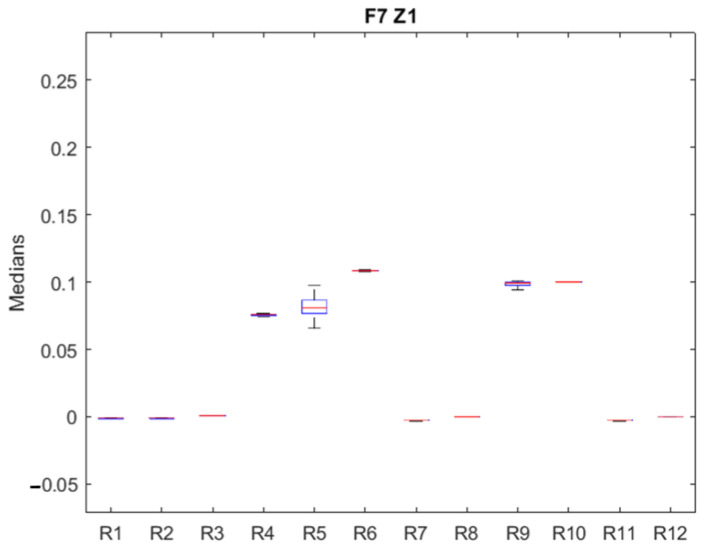
Medians of residuals for *F*_7__*Z*_1_. Medians are plotted in red, upper and lower quartile box in blue, and maximum and minimum values in black.

**Figure 8 sensors-23-07383-f008:**
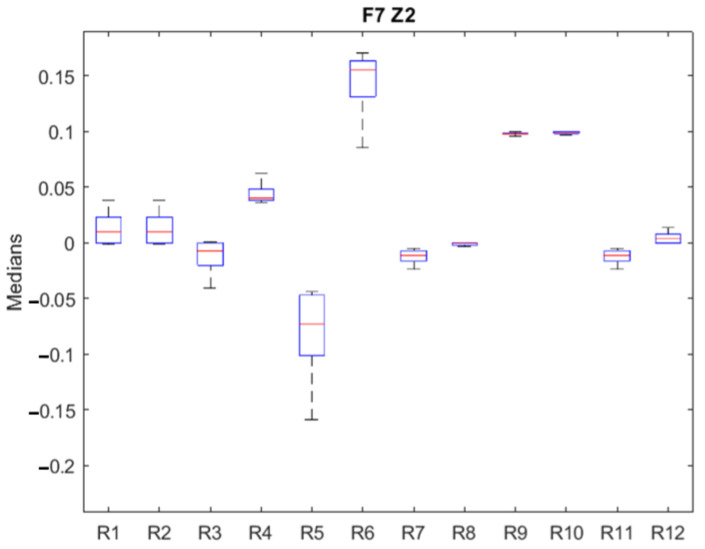
Medians of residuals for *F*_7__*Z*_2_. Medians are plotted in red, upper and lower quartile box in blue, and maximum and minimum values in black.

**Figure 9 sensors-23-07383-f009:**
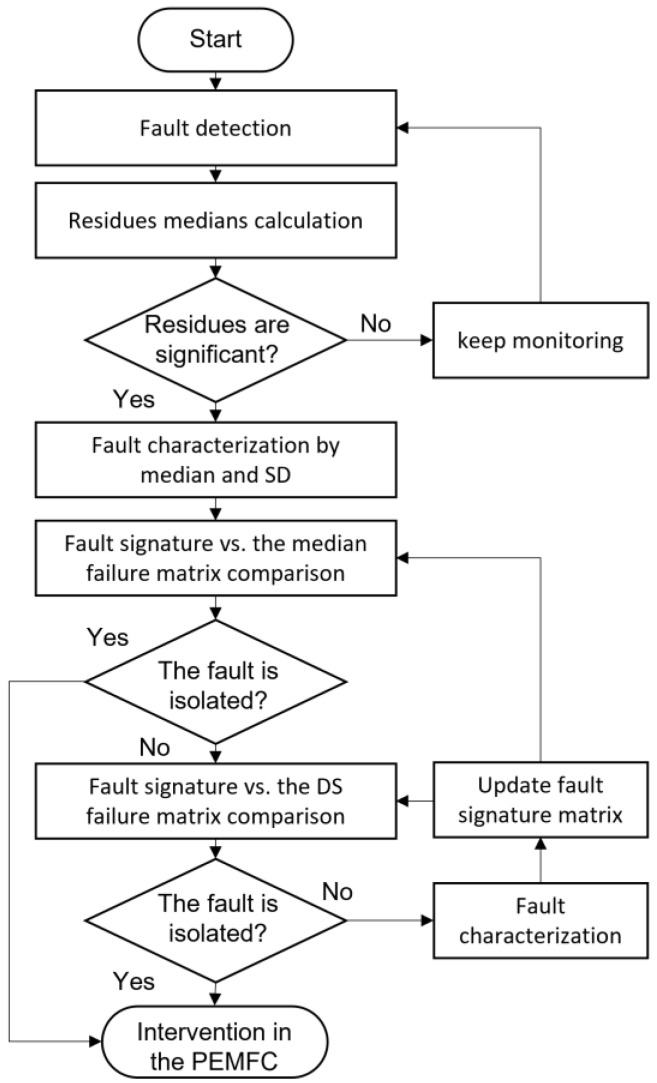
Fault diagnosis algorithm.

**Table 1 sensors-23-07383-t001:** Nexa 1.2 kW fuel cell main characteristics [29].

Signal	Value
Nominal Power	1200 W
Working voltage range	22–50 V
Maximum current	55 A
Hydrogen consumption	<18.5 slpm
Air inlet flow	90 slpm
Fuel Cell Stack Temperature	5–80 °C

**Table 3 sensors-23-07383-t003:** List of faults and generation actions.

Element	Fault Description	Simulation	Fault ID
Room temperature sensor	Sensor fault: stuck at min value	T_initial_ = 0%	F1
	Sensor fault: stuck at max value	T_initial_ = 121%	F2
Cells	Cell degradation	Cell area lowered to 80%	F3
	One cell fault	Number of cells decreased	F4
Control board	H_2_ inner pressure sensor fault	H_2__Press = 0%	F5
	O_2_ inner pressure sensor fault	O_2__Press = 0%	F6
	Current sensor calibration fault (low)	I sensor measures 10% under the real value	F7
	Current sensor calibration fault (high)	I sensor measures 10% over the real value	F8
Fuel feed	H_2_ intake press drop-leakage	H_2__Press = 80%	F9
O_2_ intake	Compressor fault	Air = 0%	F10
	Filter blockage	Air decreased by 10%	F11
	No filter or duct breakage	Air increased by 10%	F12
Air intake	Fan fault	%Fan = 0%	F13
	Blockage of the ventilation system	%Fan decreased by 20%	F14

**Table 4 sensors-23-07383-t004:** Normality test results for *F*_7__*Z*_1_.

Residual	Mean	SD	Minimum	Maximum	Shapiro-Wilk	
					W	*p*
n	−0.00126	2.97 × 10^−4^	−0.00140	1.47 × 10^−10^	0.728	<0.001
2	−0.00126	2.97 × 10^−4^	−0.00141	6.69 × 10^−9^	0.728	<0.001
3	9.09 × 10^−4^	2.11 × 10^−4^	0.00000	0.00101	0.723	<0.001
4	0.07570	0.00187	0.06269	0.07712	0.298	<0.001
5	0.08115	0.02484	−0.05517	0.26950	0.578	<0.001
6	0.10856	0.00113	0.10741	0.11638	0.318	<0.001
7	−0.00291	3.04 × 10^−4^	−0.00487	−1.68 × 10^−4^	0.416	<0.001
8	9.24 × 10^−5^	2.17 × 10^−5^	−4.35 × 10^−11^	1.03 × 10^−4^	0.725	<0.001
9	0.09929	0.00229	0.09084	0.10063	0.778	<0.001
10	0.10008	1.95 × 10^−5^	0.10000	0.10009	0.725	<0.001
11	−0.00291	3.04 × 10^−4^	−0.00487	−1.68 × 10^−4^	0.416	<0.001
12	−4.27 × 10^−4^	10.00 × 10^−5^	−4.78 × 10^−4^	−2.21 × 10^−6^	0.729	<0.001

**Table 5 sensors-23-07383-t005:** *t*-test data results for *F*_7__*Z*_1_.

Residual	*t*	Dof	*p*
1	−120	999	<0.001
2	−120	999	<0.001
3	122	999	<0.001
4	1274	999	<0.001
5	106	999	<0.001
6	3029	999	<0.001
7	−306	999	<0.001
8	121	999	<0.001
9	1357	999	<0.001
10	162,214	999	<0.001
11	−306	999	<0.001
12	121	999	<0.001

**Table 6 sensors-23-07383-t006:** Levene’s test for homogeneity of variances for *F*_7__*Z*_1_.

F	Dof	*p*
293	119,888	<0.001

**Table 7 sensors-23-07383-t007:** ANOVA Kruskal-Wallis for *F*_7__*Z*_1_.

χ^2^	Dof	*p*
293	119,888	<0.001

**Table 8 sensors-23-07383-t008:** Dwass–Steel–Critchlow–Fligner test for *F*_7__*Z*_1_.

Residuals		W	*p*	Residuals		W	*p*
1	2	−1.58	0.994	4	8	−54.76	<0.001
1	3	54.77	<0.001	4	9	54.76	<0.001
1	4	54.76	<0.001	4	10	54.81	<0.001
1	5	52.46	<0.001	4	11	−54.76	<0.001
1	6	54.76	<0.001	4	12	−54.77	<0.001
1	7	−54.18	<0.001	5	6	49.78	<0.001
1	8	54.76	<0.001	5	7	−52.46	<0.001
1	9	54.76	<0.001	5	8	−52.46	<0.001
1	10	54.81	<0.001	5	9	38.51	<0.001
1	11	−54.18	<0.001	5	10	42.31	<0.001
1	12	49.83	<0.001	5	11	−52.46	<0.001
2	3	54.77	<0.001	5	12	−52.46	<0.001
2	4	54.76	<0.001	6	7	−54.76	<0.001
2	5	52.46	<0.001	6	8	−54.76	<0.001
2	6	54.76	<0.001	6	9	−54.76	<0.001
2	7	−54.17	<0.001	6	10	−54.81	<0.001
2	8	54.76	<0.001	6	11	−54.76	<0.001
2	9	54.76	<0.001	6	12	−54.76	<0.001
2	10	54.81	<0.001	7	8	54.76	<0.001
2	11	−54.17	<0.001	7	9	54.76	<0.001
2	12	49.84	<0.001	7	10	54.81	<0.001
3	4	54.76	<0.001	7	11	0.00	1.000
3	5	52.46	<0.001	7	12	54.58	< 0.001

**Table 9 sensors-23-07383-t009:** Residuals medians of residuals for *F*_7__*Z*_1_.

Residual	Mean	Median	SD
1	−0.00113	−0.00126	2.97 × 10^−4^
2	−0.00113	−0.00126	2.97 × 10^−4^
3	8.15 × 10^−4^	9.09 × 10^−4^	2.11 × 10^−4^
4	0.07539	0.07570	0.00187
5	0.08292	0.08115	0.02484
6	0.10868	0.10856	0.00113
7	−0.00295	−0.00291	3.04 × 10^−4^
8	8.28 × 10^−5^	9.24 × 10^−5^	2.17 × 10^−5^
9	0.09826	0.09929	0.00229
10	0.10007	0.10008	1.95 × 10^−5^
11	−0.00295	−0.00291	3.04 × 10^−4^
12	−3.83 × 10^−4^	−4.27 × 10^−4^	10.00 × 10^−5^

**Table 10 sensors-23-07383-t010:** Median weighting quartiles.

Quartile	Range	Value	Quartile	Range	Value
Q1	[0.05 < x < 0.25)	1	−Q1	[−0.05 > x > −0.25)	−1
Q2	[0.25 < x < 0.50)	2	−Q2	[−0.25 > x > −0.50)	−2
Q3	[0.50 < x < 0.75)	3	−Q3	[−0.50 > x > −0.75)	−3
Q4	x > 0.75	4	−Q4	x < −0.75	−4

**Table 11 sensors-23-07383-t011:** Weighted median failure signature for F7.

Failure	Zone	R1	R2	R3	R4	R5	R6	R7	R8	R9	R10	R11	R12	Ø_Z_
F7	Z1	0	0	0	1	1	1	0	0	1	1	0	0	5
	Z2	0	0	0	0	−1	1	0	0	1	1	0	0	4
	Z3	0	0	0	0	−1	1	0	0	1	1	0	0	4
	Z4	-	-	-	-	-	-	-	-	-	-	-	-	
	Σ_r_	0	0	0	1	−1	3	0	0	3	3	0	0	

**Table 12 sensors-23-07383-t012:** Weighted standard deviation failure signature for F7.

Fault	Zone	R1	R2	R3	R4	R5	R6	R7	R8	R9	R10	R11	R12	Ø_Z_
F7	Z1	1	1	1	1	2	1	1	1	1	1	1	1	13
	Z2	2	2	2	1	2	2	1	1	1	1	1	1	17
	Z3	1	1	1	1	2	1	1	1	1	1	1	1	13
	Z4	-	-	-	-	-	-	-	-	-	-	-	-	
	Σ_r_	4	4	4	3	6	4	3	3	3	3	3	3	

**Table 13 sensors-23-07383-t013:** Intervention guidelines according to the isolated failure.

Failure	Action
F1	Check the external temperature sensor Check the environmental operating conditions of the module
F2	
F3	Initiate cell rejuvenation procedure Periodic parametric identification
F4	Inspection of the gas diffusion channels of the stack Inspection of cell connections Initiate cell rejuvenation procedure
F5	Hydrogen pressure sensor check Hydrogen availability check Hydrogen supply system check
F6	Oxygen sensor check Oxygen supply system check
F7	Current sensor check
F8	Control board check
F9	Hydrogen sensor check Control board check
F10	Compressor overhaul Compressor electrical connections check Mass airflow sensor overhaul
F11	Compressor filter overhaul Review of air supply and exhaust ducts
F12	Compressor filter overhaul Duct check
F13	Overhaul of the fan motor Review of fan electrical connections.
F14	Review of fan fastening Filter overhaul Review of ducts

## Data Availability

The data presented in this study are available on request from the corresponding author.

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
