# Peer review of "PEMFCs Model-Based Fault Diagnosis: A Proposal Based on Virtual and Real Sensors Data Fusion"

_sensors, 2023, doi:10.3390/s23177383_

Round 1

Reviewer 1 Report

In this manuscript, authors proposed a model-based fault diagnosis of PEMFCs based on virtual and real sensors data fusion, which is a good work. I would like to recommend acceptance of this manuscript after several minor revisions:

1.      In the Abstract, ‘the research was conducted in the … Spain’ is recommended to be deleted in the Abstract;

2.      Page 2, ‘Water and temperature management’ should be ‘water and thermal management’;

3.      Page 2, ‘through the bipolar plates and diffusion channels’, what does the ‘diffusion channels’ refer?

4.      Page 2, ‘gas exchange between the cathode and anode channels’, how?

Minor improvement is recommended. 

Author Response

We would like to thank you for your valuable comments, and we appreciate the time you spent reviewing our work. We have addressed your comments in the attached document.

Reviewer 2 Report

This paper presents a novel algorithm for detecting and isolating primary faults in Proton Exchange Membrane Fuel Cells (PEMFCs) by combining virtual and real sensor data fusion. The algorithm is tested on a PEMFC system and is shown to accurately detect and isolate faults. The paper also discusses the benefits of reliable fault diagnosis in renewable hybrid systems that use PEMFCs. However, there are some problems, such as references, the explanation of the principle and grammar, etc. Detailed comments are listed as follows. Hope they can provide help for improving the quality of this paper. 

1) The references to this article are insufficient. The following references are recommended to be added.

(1) Jian Yu, Yu Wen, Lei Yang*, Zhibin Zhao**, Yanjie Guo, Xiao Guo, Monitoring on Triboelectric Nanogenerator and Deep Learning Method, Nano Energy, 2022, 92, 106698.

2)How does the proposed algorithm compare to other existing fault diagnosis algorithms for PEMFCs?

3)What are the specific primary faults that the proposed algorithm can detect and isolate in PEMFCs?

 4)How does the virtual sensor data fusion work in the proposed algorithm, and what are the benefits of using virtual sensors?

 5)How was the model for the PEMFC system developed, and what were the key parameters used in the model?

 6)What are the limitations of the proposed algorithm, and how can they be addressed in future research?

Author Response

(The authors gave the same response as above.)

Reviewer 3 Report

1. Author mentioned "Algorithm achieved an error of 2.21% in voltage and 1.97% in temperature" Can you provide a graph to show this? Compare algorithm data with available data.

2. Which software was used in developing the algorithm?

3. What is the difference between warning alerts and fault alerts?

4. How did the author make sure that one of the models runs fault-free?

5. What are Kolmogorov-Smirnov test and Shapiro-Wilk test? It will be confusing for the first-time readers. Please cite them or give more details in appendix. 

6. Same with the test the author is using in the model Kruskal-Wallis test. Please give some details.

7. From author's research which fault is most likely to occur and why? 

8. Why and how does the author feel this work is important? Add couple of lines in conclusion

Author Response

First, we would like to thank you for your valuable comments, and we appreciate the time you spent reviewing our work. We have addressed your comments as follows:

  1. Author mentioned "Algorithm achieved an error of 2.21% in voltage and 1.97% in temperature" Can you provide a graph to show this? Compare algorithm data with available data.

Response: Thank you for your comment. The graphs comparing the actual voltage and temperature values with those obtained from the model are now shown in Figure 3.

  1. Which software was used in developing the algorithm?

Response: Thank you for your comment. To develop the model, LabView was used. It is now mentioned in the abstract, in section 2.2, and in the conclusions.

  1. What is the difference between warning alerts and fault alerts?

Response: Thank you for your comment. Warning alerts indicate potential issues that require attention but don't disrupt the immediate operation. Fault alerts signify critical malfunctions that demand immediate action to prevent system failure. This information has been added to section 2.3 PEMFC faults.

  1. How did the author make sure that one of the models runs fault-free?

Response: Thank you for your comment.

The model was fitted with fault-free real data from the fuel cell device. Therefore, this model, without any modification, can be considered fault-free.

  1. What are Kolmogorov-Smirnov test and Shapiro-Wilk test? It will be confusing for the first-time readers. Please cite them or give more details in appendix.

Response: Thank you for your comment.

The Kolmogorov-Smirnov and Shapiro-Wilk tests are statistical methods used to assess the normality of a dataset. The Kolmogorov-Smirnov test compares the cumulative distribution of the data with a theoretical normal distribution, yielding a single statistic. The Shapiro-Wilk test calculates a statistic based on the correlation between the data and the expected normal distribution. Both tests provide a p-value that indicates whether the data significantly deviates from a normal distribution. A low p-value suggests non-normality. While the Kolmogorov-Smirnov test is sensitive to deviations throughout the distribution (it is used for n≥50 samples), the Shapiro-Wilk test is particularly effective for smaller sample sizes (n<50 samples).

Determining whether the analyzed residuals are normal is a crucial step in the fault detection. This is achieved by applying a T-student test, which determines whether the mean of the analyzed data differs from "zero."

This information has been added to section 3. Fault diagnosis process. References have been added.

  1. Same with the test the author is using in the model Kruskal-Wallis test. Please give some details.

Response: Thank you for your comment. The Kruskal-Wallis test is a non-parametric statistical test used to determine if there are significant differences in the medians of three or more independent groups. It's an extension of the one-way ANOVA for situations where data doesn't meet the assumptions of normality or homogeneity of variances. This test determines differences between residuals pairwise to establish the fault signature.

Same with the last comment, this information has been added to section 3. Fault diagnosis process. A reference has been added.

  1. From author's research which fault is most likely to occur and why?

Response: Thank you for your comment. According to the conducted tests, the most likely failures on the PEMFC are related to temperature and water within the module, meaning membrane flooding or dehydration. In the second place, membrane-related failures can be mentioned, either due to degradation or cracks. However, common failures result from a direct fault in some devices, such as the fan, pressure regulators, gas flow sensors, etc. This information has been added to the conclusion section.

  1. Why and how does the author feel this work is important? Add couple of lines in conclusion

Response: Thank you for your comment. The importance of the work developed is now explained in the conclusion section.

Round 2

Reviewer 2 Report

The authors have addressed the issues I raised previously to my satisfaction, and I have no other questions. I recommend the paper to be accepted.